# LncMirNet: Predicting LncRNA–miRNA Interaction Based on Deep Learning of Ribonucleic Acid Sequences

**DOI:** 10.3390/molecules25194372

**Published:** 2020-09-23

**Authors:** Sen Yang, Yan Wang, Yu Lin, Dan Shao, Kai He, Lan Huang

**Affiliations:** 1Key Laboratory of Symbol Computation and Knowledge Engineering of Ministry of Education, College of Computer Science and Technology, Jilin University, Changchun 130012, China; yangsen18@mails.jlu.edu.cn (S.Y.); shaodan17@mails.jlu.edu.cn (D.S.); hekai18@mails.jlu.edu.cn (K.H.); huanglan@jlu.edu.cn (L.H.); 2School of Artificial Intelligence, Jilin University, Changchun 130012, China; linyu19@mails.jlu.edu.cn

**Keywords:** LncRNA–miRNA interactions, RNA sequence features, deep learning, computational frame

## Abstract

Long non-coding RNA (LncRNA) and microRNA (miRNA) are both non-coding RNAs that play significant regulatory roles in many life processes. There is cumulating evidence showing that the interaction patterns between lncRNAs and miRNAs are highly related to cancer development, gene regulation, cellular metabolic process, etc. Contemporaneously, with the rapid development of RNA sequence technology, numerous novel lncRNAs and miRNAs have been found, which might help to explore novel regulated patterns. However, the increasing unknown interactions between lncRNAs and miRNAs may hinder finding the novel regulated pattern, and wet experiments to identify the potential interaction are costly and time-consuming. Furthermore, few computational tools are available for predicting lncRNA–miRNA interaction based on a sequential level. In this paper, we propose a hybrid sequence feature-based model, LncMirNet (lncRNA–miRNA interactions network), to predict lncRNA–miRNA interactions via deep convolutional neural networks (CNN). First, four categories of sequence-based features are introduced to encode lncRNA/miRNA sequences including k-mer (k = 1, 2, 3, 4), composition transition distribution (CTD), doc2vec, and graph embedding features. Then, to fit the CNN learning pattern, a histogram-dd method is incorporated to fuse multiple types of features into a matrix. Finally, LncMirNet attained excellent performance in comparison with six other state-of-the-art methods on a real dataset collected from lncRNASNP2 via five-fold cross validation. LncMirNet increased accuracy and area under curve (AUC) by more than 3%, respectively, over that of the other tools, and improved the Matthews correlation coefficient (MCC) by more than 6%. These results show that LncMirNet can obtain high confidence in predicting potential interactions between lncRNAs and miRNAs.

## 1. Introduction

Although noncoding RNAs (ncRNAs) [1] cannot encode proteins, they play indispensable roles in numerous life processes [2,3,4,5,6,7]. Accumulated studies show that many ncRNAs are involved in various life regulation processes [8,9]. LncRNA and miRNA, as two typical ncRNAs, are proof related to cancer development, gene regulation, cellular metabolic process, etc. miRNA is a small ncRNA with 20–25 nt adhering to lncRNA (more than 200 nt) to indirectly regulate gene expression [5], adjust lncRNA function, and cooperate with lncRNA to finish regulation processes. The increasing evidence shows that the interaction between lncRNA and miRNA contributes to finding some potential regulation relationships. Therefore, exploring lncRNA–miRNA interactions can support the understanding of some of the complicated functions between lncRNAs and miRNAs. In earlier studies, researchers mainly explored unknown lncRNA–miRNA interactions through laboratory experiments. However, finding potential interaction between lncRNAs and miRNAs by a biological laboratory is labor-intensive, time-consuming, and costly. Meanwhile, with the rapid development of RNA sequencing technology, a fair number of novel lncRNAs and miRNAs have been detected. Hence, many computational methods to predict lncRNA–miRNA interactions have been proposed. In 2018, Huang et al. introduced a group preference Bayesian collaborative filtering model (GBCF) for picking up a top-k probability ranking list for an individual miRNA or lncRNA based on the known miRNA–lncRNA interaction network [10]. In the same year, Huang et al. employed a graph-based prediction method (Expression Profile-based prediction model for LncRNA-MiRNA Interactions, EPLMI) to infer the most potential lncRNA–miRNA interactions based on the known lncRNA–miRNA interaction network, lncRNA–lncRNA similarity, and miRNA–miRNA similarity [11]. In 2019, Huang et al. proposed a graph convolution auto-encoder network method that incorporated the raw data of node attributes and topology of the interaction network to predict the link between lncRNA and miRNA [12]. With the rapid development of graph embedding technology, in 2019, Zhou et al. proposed an ensemble graph embedding method (GEEL) [13] that used linear neighborhood similarity (LNS) method and known interactions to construct a lncRNA–miRNA interaction graph, and then introduced four graph embedding methods (Laplacian Eigenmaps, GraRep, High Order Proximity preserved Embedding, DeepWalk) and a graph auto encoder model to represent the lncRNA/miRNA node. Based on the embedding results, GEEL used the random forest classifier to predict the potential interactions between lncRNAs and miRNAs. In 2020, Kang et al. used raw RNA sequence and 110 sequence-based features to feed a BiGRU model and a random forest model, respectively, to train a hybrid model (PmliPred) for predicting the potential interactions between lncRNAs and miRNAs [14]. Although the above methods can relieve some problems, they also have some limitations. For example, EPLMI needs the expression of lncRNAs and miRNAs as input, but it is difficult to obtain the special expression of lncRNAs and miRNAs most times. Additionally, the expression of lncRNAs and miRNAs are tissue specific and inconsistent between different quantitative methods. Another popular method, GBCF, can predict potential interaction between known lncRNA and miRNA, while it is difficult to predict the interaction between novel lncRNAs and miRNAs. Additionally, PmliPred is an expert in predicting the interaction between lncRNAs and miRNAs in plants, while it performs poorly in animals. In order to solve these limitations, we propose a sequence features-based model called LncMirNet. LncMirNet introduces RNA sequence-based features including k-mer features and composition transition distribution (CTD) features [15] as well as deep learning-based features including doc2vec and graph embedding as input properties. LncMirNet employed the convolutional neural networks (CNN) model to predict potential interactions between lncRNAs and miRNAs. It contains three main steps: (1) constructing k-mer, CTD, doc2vec, and graph embedding features; (2) using the histogram-dd method to convert the constructed lncRNA/miRNA sequence features into their corresponding matrix, respectively; and (3) employing the CNN model to relearn the above constructing matrix features to predict the potential interactions between lncRNAs and miRNAs. For the public benchmark datasets (lncRNASNP2) [16], LncMirNet successfully predicted the potential interactions between lncRNAs and miRNAs with high evaluation performances. Compared with the other six state-of-the-art methods by five-fold validation, LncMirNet achieved more than 3% higher accuracy and AUC, respectively. Specifically, in terms of MCC, LncMirNet obtained over 6% improvement compared with other methods. Furthermore, LncMirNet outperformed all the competing methods on other metrics. From the experimental results, we can conclude that LncMirNet is a robust and high confidence tool for predicting the potential interactions between lncRNAs and miRNAs with RNA sequence-based features alone.

## 2. Materials and Methods

### 2.1. Materials

#### 2.1.1. Datasets

The positive training data were obtained from the lncRNASNP2 database (January 2018 version) [16], which is available at http://bioinfo.life.hust.edu.cn/. In the lncRNASNP2 database, the true interactions were confirmed by laboratory examinations with the research literature. lncRNAs in lncRNASNP2 are indicated by Ensemble ID and we downloaded the corresponding human lncRNA sequences from GENCODE (https://www.gencodegenes.org/) [17]. We also extracted human miRNA sequences from the miRbase database (http://www.mirbase.org/) [18]. To filter out true interactions, we selected positive lncRNA–miRNA pairs when a record in lncRNASNP2 appeared as ‘hsa-miR’ and ‘ENST’ simultaneously. Finally, we obtained 258 miRNAs, 1663 lncRNAs, and 15,386 validation lncRNA–miRNA interactions and their corresponding sequences.

#### 2.1.2. Constructing Positive and Negative Samples

In the lncRNASNP2 database, there were 15,386 validation lncRNA–miRNA interactions treated as positive samples. For negative interactions, we applied the same strategy widely used in the previous research to construct negative interactions including GEEL, SG-LSTM [19], and GCLMI. First, the Knuth–Durstenfeld shuffle algorithm [20] was utilized to shuffle the lncRNA set and miRNA set 10 times, respectively, and then a lncRNA (as lncRN1) and a miRNA (as miRNA1) were randomly selected from the lncRNA set and miRNA set individually. Second, if the lncRNA1–miRNA1 pair did not appear in the positive interactions and negative interactions, the lncRNA1–miRNA1 interaction was regarded as a negative sample. Finally, for the balance of true and negative samples, we repeated the shuffle and selection process and obtained 15,386 negative samples.

### 2.2. Methods

#### 2.2.1. Overall Workflow

In our method, k-mer [21], CTD [15], and doc2vec [22] features were computed first for the lncRNA/miRNA sequences. Then, based on these features, the linear neighborhood similarity measure (LNS) [13] was applied to construct a lncRNA/miRNA neighborhood graph. After that, role2vec [23], a graph embedding method, was employed to embed each node. Role2vec incorporates both the graph structure and node attribute information to learn the representation for each node. Sequentially, the k-mer, CTD, doc2ve, and graph embedding features of lncRNAs/miRNAs were fused to a matrix for fitting the CNN learning pattern by a Histogram-dd. Histogram-dd can fuse multiple vectors to a histogram matrix. Finally, histogram matrices are fed to a CNN model. The CNN model uses filters to distill deep features to learn how to predict the potential interactions between lncRNAs and miRNAs. The overall workflow of LncMirNet is shown in Figure 1.

#### 2.2.2. Construction Features

##### k-mer Features of RNA Sequence

The RNA sequence consists of Adenine (A), Uracil (U), Cytosine (C), and Guanine (G). In this paper, Uracil (U) in the RNA sequence was replaced by T (Thymidine). For a RNA sequence, the k-mer frequency distribution is a basic and indispensable feature that can be represented by the k-mer frequency. Four kinds of k-mer features including 1-mer, 2-mer, 3-mer, and 4-mer are introduced, where 1-mer records the counts of A, T, C, G; 2-mer saves the frequencies of AA, AT, …, GG; 3-mer holds the times of AAA, AAT, …, GGG; and 4-mer stores the numbers of AAAA, AAAT, …, GGGG. Finally, four kinds of k-mer features are merged into a vector with 340 ($4^{1}+4^{2}+4^{3}+4^{4}=340$) dimensions in all. Notedly, in this paper, for a miRNA sequence, we only computed 1-mer, 2-mer, and 3-mer features since the miRNA sequence is usually short (average length less than 30 nt) and 4-mer features for a miRNA are usually sparse.

##### Composition/Transition/Distribution (CTD) Features

Composition transition distribution (CTD) [1] is primarily proposed for predicting the protein folding class, which is a global protein sequence descriptor established by Dubchak’s work [24]. Lately, CTD features are found to relate to RNA structure and are seldom used to predict the interactions between lncRNAs and miRNAs. Therefore, in this paper, we applied CTD features to represent RNA structure information. CTD features with 30 dimensions are sourced from Composition, Transition, and Distribution, where the Composition features are the number of amino acids of a particular property divided by the total number of amino acids, the Transition features characterize the percent frequency with which amino acids of a particular property are followed by amino acids of different property, and the Distribution features measure the chain length within which the first, 25%, 50%, 75%, and 100% of the amino acids of a particular property are located.

For example, we used a toy RNA sequence ATACGTACTGCT GACGTAGC to show how to calculate the CTD features. The toy RNA sequence contains 5 A, 5 T, 5 G, and 5 C, so the composition is equal to 5/20 = 0.25, 5/20 = 0.25, 5/20 = 0.25, and 5/20 = 0.25. Transition includes AT, AC, AG, TG, TC, and GC, six features that describe the percent frequency with the conversion of four nucleotides between adjacent positions. AT represents the percent frequency of A adjoining T or T adjoining A. AC, AG, TG, TC, and GC are the same formulation of AT. Therefore, the transition for the toy RNA sequence is equal to 2/19 = 0.105, 3/19 = 0.158, 2/19 = 0.105, 4/19 = 0.211, 2/19 = 0.105, and 4/19 = 0.211. Distribution is five relative positions along the transcript sequence of each nucleotide, with 0 (first node), 25, 50, 75, and 100% (last node) to measure the nucleotide distribution. For As, the 0% was located at the first position in the toy RNA sequence, 25, 50, 75, and 100% at the third, seventh, fourteenth, and eighteenth position, respectively. So, As was 1/20 = 0.05, 3/20 = 0.15, 7/20 = 0.35, 14/20 = 0.7, and 18/20 = 0.9. Likewise, Ts, Gs, and Cs were 0.1, 0.3, 0.45, 0.6, 0.85, 0.25, 0.5, 0.65, 0.8, 0.95, 0.2, 0.4, 0.55, 0.75, and 1. We used A0, A1, A2, A3, A4, T0, T1, T2, T3, T4, G0, G1, G2, G3, G4, C0, C1, C2, C3, and C4 to represent the 20 features [1].

##### Distributed Representation Feature of RNA Sequence by doc2vec

A RNA sequence can be regarded as a sentence. Therefore, encoding sentence methods in neural language process (NLP) [25] can be introduced to represent RNA sequences. In this paper, doc2vec [22] was recommended to construct the distributed representation feature for a RNA sequence. Doc2vec uses local context and sentence global information to learn sentence representation. First, a continuous RNA sequence is segmented by a 3-mer window with forwarding step 1. Second, the segmented 3-mers are applied to train a doc2vec model. Finally, based on the trained doc2vec, any RNA sequence can be encoded into a fixed-size vector. The pipeline of RNA sequence encoded by doc2vec is shown in Figure 2 where the sequence global information records the sequence index that will be inferred to a fixed size vector to represent the RNA sequence.

##### Graph Embedding Methods to Represent RNA Sequence

The lncRNA–lncRNA/miRNA–miRNA interaction graph contains graph structure information. The graph information contributes to encoding the lncRNA/miRNA sequence. Each node in the graph indicates a lncRNA/miRNA and each edge illustrates their interaction. To construct the lncRNA/miRNA interactions graph, the k-mer count, CTD, and doc2vec encoding feature of a lncRNA/miRNA sequence were merged into union vectors. Sequentially, the union vectors were utilized to construct the lncRNA–lncRNA similarity matrix by LNS. For example, a lncRNA (as lnc1) whose top 15 close distance lncRNAs with a similarity weight larger than 0 is thought to be existing connections. Based on this strategy, the closely homologous lncRNAs are linked to Lnc1 to build the lncRNA–lncRNA interaction graph. For the miRNA–miRNA interaction graph, the building process was similar to the construction of the lncRNA–lncRNA interaction graph. Finally, the role2vec embedding method was employed to encode each node since role2vec can fully use graph structure and node attributes. In this paper, the embedding dimension was set as 128, the random walk order was set as one, and the rest of the parameters were set as the value based on author suggestion. After the graph embedding process, both lncRNA and miRNA were represented by 128 dimensional-vector fusing sequence and geometric information.

##### Constructing Matrix Features by Histogram-Dd

The strategy of fusing multiple category features may support the improvement of the performance of a classifier. Therefore, we used the histogram-dd method to convert lncRNA/miRNA vectors into their corresponding matrices, which exactly fit the CNN learning pattern. Like the approaches used in [26], a lncRNA/miRNA sequence can be disintegrated into three sub-sequences by starting at positions 0, 1, and 2, respectively. For these sub-sequences of lncRNA/miRNA, each sub-sequence was used to compute their corresponding k-mer, CTD, doc2vec, and graph embedding features. Finally, histogram-dd integrated these four categories of features of lncRNA/miRNA into their corresponding matrices with a size of 20×20×4, respectively. The advantage of converting the lncRNA/miRNA vectors into matrices is that the transformed matrices not only save the original information of one-dimensional features but also fit the CNN learning pattern that supports the interaction prediction between lncRNAs and miRNAs.

#### 2.2.3. Prediction Model by Convolutional Neural Networks

Deep learning technology has obtained numerous achievements in many bioinformatics applications. As one of the important deep learning models, CNN applies convolutional kernels to automatically extract potential features from the raw input data matrix. Many successful bioinformatics applications have proven that CNN is a powerful algorithm to solve classification and regulation problems. Hence, CNN is employed to predict the interactions between lncRNAs and miRNAs. The CNN predictor model consists of multiple convolution layers, dense layers (fully connected layers), batch normalization layers, dropout layers, etc. First, the inputs of the CNN model are two tensors with size a 20×20×4 corresponding to lncRNA and miRNA. After traversing multiple CNN layers, respectively, lncRNA tensors and miRNA tensors are merged as a fusing tensor for connecting dense layers. Each convolution layer consists of multiple filters with a 3×3 kernel size, stride one, and Rule activation function. Dropout layers are embedded into convolution layers to enhance the robustness of CNN. Batch normalization layers are employed to normalize the intermediate data to accelerate better training. We selected a sigmoid activation function on the output layer. When the predicted result was larger than 0.5, we believe that the candidate lncRNA–miRNA pair interaction exists. The detailed structure and parameters of LncMirNet are shown in Appendix A.

### 2.3. Implementation of LncMirNet

LncMirNet was implemented by Keras 2.3.1 with backend Tensorflow 1.15.0 and all scripts were written by Python 3.6. LncMirNet was run on a PC with 4.3 GHz, 8 cores CPU, and 16 GB RAM under an open Linux operating system.

### 2.4. Evaluation Criteria

LncMirNet was evaluated by the widely used standard performance metrics, which are sensitivity (SN), specificity (SP), accuracy (ACC), F1-score ($F_{1}$), and Matthews correlation coefficient (MCC). These evaluation metrics are defined as follows:(1)$$\mathrm{Sensitivity}\left(\mathrm{SN}\right)=\frac{\mathrm{TP}}{\mathrm{TP}+\mathrm{FN}}$$
(2)$$\mathrm{Specificity}\left(\mathrm{SP}\right)=\frac{\mathrm{TN}}{\mathrm{TN}+\mathrm{FP}}$$
(3)$$\mathrm{Accuracy}\left(\mathrm{ACC}\right)=\frac{\mathrm{TP}+\mathrm{TN}}{\mathrm{TP}+\mathrm{TN}+\mathrm{FP}+\mathrm{FN}}$$
(4)$$F1-\mathrm{score}\left(F_{1}\right)=\frac{2\mathrm{TP}}{2\mathrm{TP}+\mathrm{FP}+\mathrm{FN}}$$
(5)$$\mathrm{Mattews}\;\mathrm{correlation}\;\mathrm{coefficient}\;\left(\mathrm{MCC}\right)\;=\frac{\mathrm{TP}\times \mathrm{TN}-\mathrm{FP}\times \mathrm{FN}}{\sqrt{\left(\mathrm{TP}+\mathrm{FN}\right)\times \left(\mathrm{TP}+\mathrm{FP}\right)\times \left(\mathrm{TN}+\mathrm{FP}\right)\times \left(\mathrm{TN}+\mathrm{FN}\right)}}$$
where TP, FP, TN, and FN represent the true positives, false positives, true negatives, and false negatives, respectively. We also plotted the receiver operating characteristic curves (ROC) and computed the area under the curve (AUC) to precisely show the different performances of each model.

## 3. Results

### 3.1. Experimental Settings

When constructing the k-mer features, 1-mer (4^1^ = 4), 2-mer (4^2^ = 16), 3-mer (4^3^ = 64), and 4-mer (4^4^ = 256) with forwarding step one were used, so altogether, 340 features were generated to represent a lncRNA sequence, while only 1-mer, 2-mer, and 3-mer were used for a miRNA sequence due to the short length of miRNA. The CTD method produces a 30-dimensional feature to encode a lncRNA/miRNA sequence. Doc2vec is an unsupervised method that combines local context information and sequence global information of a RNA sequence to indicate any RNA length sequence to a fixed size vector. Before training a doc2vec model, a lncRNA/miRNA sequence will be segmented into 3-mer items. Based on the segmented 3-mer items of a lncRNA/miRNA sequence, the distributed memory (PV-DM) [27] strategy is utilized to train a doc2vec model. We set a 128-dimensional vector and a 64-dimensional vector to hold the sequence global information of a lncRNA sequence and a miRNA sequence, respectively. For the graph embedding method, we employed the role2vec method. Role2vec is utilized to generate embedding expressions that focus on both the structure and neighbor information of networks. Each node in the lncRNA/miRNA graph is embedded into a 128-dimensional vector. After the process of feature construction, we obtain a 340-dimensional k-mer vector, a 30-dimensional CTD vector, a 128-dimensional doc2vec vector, and a 128-dimensional neighbor graph embedding vector for a lncRNA sequence as well as an 84-dimensional k-mer vector, a 30-dimensional CTD vector, a 64-dimensional doc2vec vector, and a 128-dimensional neighbor graph embedding vector for a miRNA sequence. Due to the CNN learning pattern, being friendly with the matrix data as input, we adopted a histogram-dd method to convert lncRNA/miRNA vectors into a 20×20×4 matrix, respectively, and fed these matrices to a CNN model for training.

### 3.2. The Effects of Feature Combination

To explore the performance of different combinations of four types of characteristics (k-mer features, CTD, doc2vec, graph embedding features), a five-fold cross validation experiment was conducted on all data. As shown in Table 1, on the training and test datasets, LncMirNet trained with all kinds of features achieved better performance and obtained the lowest accuracy when it used the k-mer features alone. The results show that the integration of four types of characteristics is a compelling combination for predicting the potential interactions between lncRNAs and miRNAs. Table 1 records the five-fold cross validation results.

### 3.3. Comparison with Six Other Methods on All Data

To evaluate the performance of LncMirNet, LncMirNet was compared with the other six state-of-the-art methods including GEEL, PmliPred, BiLSTM, SEAL, SVD, and Katz. GEEL calculated the lncRNA–lncRNA/miRNA–miRNA similarity matrix by the linear neighborhood similarity method by 5-mer features and constructed the lncRNA/miRNA interaction heterogeneous network. GEEL uses multiple graph embedding methods and a graph auto encoder [28] to represent each lncRNA/miRNA sequence and trains a random forest classifier to predict feasible interactions. PmliPred is based on a hybrid model and fuzzy decision for plant lncRNA–miRNA interactions prediction. PmliPred approves 110 features including k-mer frequency, GC content, base pairs number, and minimum free energy of lncRNA/miRNA sequences to form features and trains a random forest model. Furthermore, PmliPred also encodes the RNA sequence by one-hot and trains a CNN-BiGRU model. Based on the predicted result of the random forest model and CNN-BiGRU model, PmliPred uses the fuzzy decision to predict the final result. LncRNA–miRNA interaction problems can be regarded as a graph link prediction problem. In this paper, we also introduced a link prediction model, namely Subgraphs, Embeddings and Attributes for Link prediction (SEAL), as a compared method. SEAL learns heuristics from local subgraphs using a graph neural network (GNN), which is enabled to obtain better graph feature learning ability. BiLSTM, a time-series deep learning model, was also introduced as a compared method since the RNA sequence is a time-series data struct. BiLSTM is fed by one-hot embedding matrix of RNA sequence and outputs the predicted interaction probability. Additionally, we also selected Singular Value Decomposition (SVD), a traditional matrix factorization method, as a compared method, which uses the decomposition method of adjacency matrix to represent lncRNA and miRNA. Based on the representation of SVD, the random forest model was used to classify the interaction between lncRNA and miRNA. Katz is often used to link prediction problems, which can distinguish the influence of different neighbor nodes and get the influence value of each node. Based on the node influence, Katz utilizes the random forest model to predict the potential interactions between lncRNAs and miRNAs.

Five-fold cross validation was used to evaluate the performance of seven models. All lncRNA–miRNA interactions were randomly divided into five equal-sized subsets. Furthermore, Sensitivity, Specificity, F1-score, Accuracy, and MCC were adopted as evaluation metrics and AUCs were calculated and ROCs plotted to distinguish the performance of each prediction model. As shown in Table 2, LncMirNet achieved a MCC score of 0.7124 and AUC score of 0.9381, which outperformed GEEL (MCC score: 0.6445; AUC score: 0.8982), PmliPred (MCC score: 0.6004; AUC score: 0.9030), BiLSTM (MCC score: 0.4359; AUC score: 0.7876), SEAL (MCC score: 0.5754; AUC score: 0.8658), SVD (MCC score: 0.3142; AUC score: 0.7156), and Katz (MCC score: 0.1930; AUC score: 0.6459). On the other metrics including Sensitivity, Specificity, F1-score and Accuracy, LncMirNet outperformed the compared methods most times. The superior performances of LncMirNet are due to two reasons. On one hand, LncMirNet fully exploits the structure information of the lncRNA–lncRNA/miRNA–miRNA graph by graph embedding methods and integrates multiple RNA sequence features including k-mer, CTD, and doc2vec features. On the other hand, LncMirNet converts lncRNA/miRNA vectors to a matrix by histogram-dd and employs a powerful CNN model to relearn potential features for improving the performance of LncMirNet. Based on these superiorities, LncMirNet obtained better performance compared with the other state-of-the-art methods. Moreover, LncMirNet is not limited to predict the interactions between known lncRNAs and known miRNAs. The learning pattern of LncMirNet also fits other interaction problems. We also plotted the ROC curve of seven methods to further show the distinct performance. Figure 3 shows the seven ROC curves, and we can see that the LncMirNet curve was above on all compared methods.

### 3.4. Negative Samples Analysis

In this paper, the pairs between lncRNAs and miRNAs without interaction were treated as negative samples. We generated the same number of negative samples as known positive samples to obtain a balanced dataset. However, positive datasets and negative datasets are not balanced most of time. Therefore, we tried to explore the performance of LncMirNet in the different sample ratios. β records the different ratios between negative samples and positive samples and ranged from 0.25, 0.5, 1.0, 2.0 to 4.0. Table 3 shows the experimental results. Although the dataset was not balanced, LncMirNet could still obtain excellent results. When the ratio β was 1.0, LncMirNet had a higher AUC value. The unbalanced dataset experiments showed that LncMirNet is a robust and reliable model to predict potential interactions between lncRNAs and miRNAs.

## 4. Discussion

Deep learning technology has yielded inspiring achievements for many bioinformatics issues [29]. Due to the increasing training data and relatively complex network struct, the issue of identifying lncRNA–miRNA interaction is a significant and indispensable step in exploring the functions between lncRNA and miRNA. With the increasing development of RNA sequence technology, numerous novel lncRNA and miRNA have been found. How to determine their interactions easily is of utmost urgency. In this paper, a hybrid features-based deep learning model was proposed. First, k-mer, CTD, and doc2vec features were utilized to represent lncRNA/miRNA sequences. Then, based on the sequence features of lncRNAs/miRNAs, a lncRNA-lncRNA graph and a miRNA–miRNA graph were generated by LNS. To fully distill the graph information, role2vec, a graph embedding method, was introduced to indicate each node (lncRNA/miRNA). The hybrid features can fully encode a lncRNA/miRNA sequence from different perspectives. Sequentially, lncRNA/miRNA features were converted to their corresponding matrices by histogram-dd for feeding a CNN model. These converted matrices not only hold original information, but also fit a CNN learning pattern. These pipelines are homologous to cascade learning with twice-feature learning. Additionally, the learning pattern can be extended to other similar problems such as protein–protein interaction, gene–miRNA interaction, etc. Therefore, due to the hybrid features embedding and CNN learning patterns, LncMirNet achieved a superior performance.

## 5. Conclusions

In this paper, we proposed a novel method based on hybrid sequence features including k-mer, CTD, doc2vec, and graph embedding features and a CNN model, namely LncMirNet, to predict lncRNA–miRNA interactions. LncMirNet is an efficient method that only relies on RNA sequence-based features. The compared experiments on lncRNASNP2 show that LncMirNet achieved more than 3%, a 6% improvement in terms of AUC and MCC, respectively, and has good generalization ability on unbalanced datasets. The predicted results of LncMirNet may lay the foundation for the construction of the lncRNA–miRNA interaction database in the future. LncMirNet may also provide valuable references for other interaction prediction studies. Overall, LncMirNet successfully identified lncRNA–miRNA interactions by sequence features alone and may contribute to other interaction research.

## 6. Data Availability Statement

Publicly available datasets were analyzed in this study. Codes and data are available at https://github.com/abcair/LncMirNet, which contains detailed steps to run LncMirNet. Moreover, a Appendix A is available to illustrate how to predict interactions between novel lncRNAs and miRNAs.

## Figures and Tables

**Figure 1 molecules-25-04372-f001:**
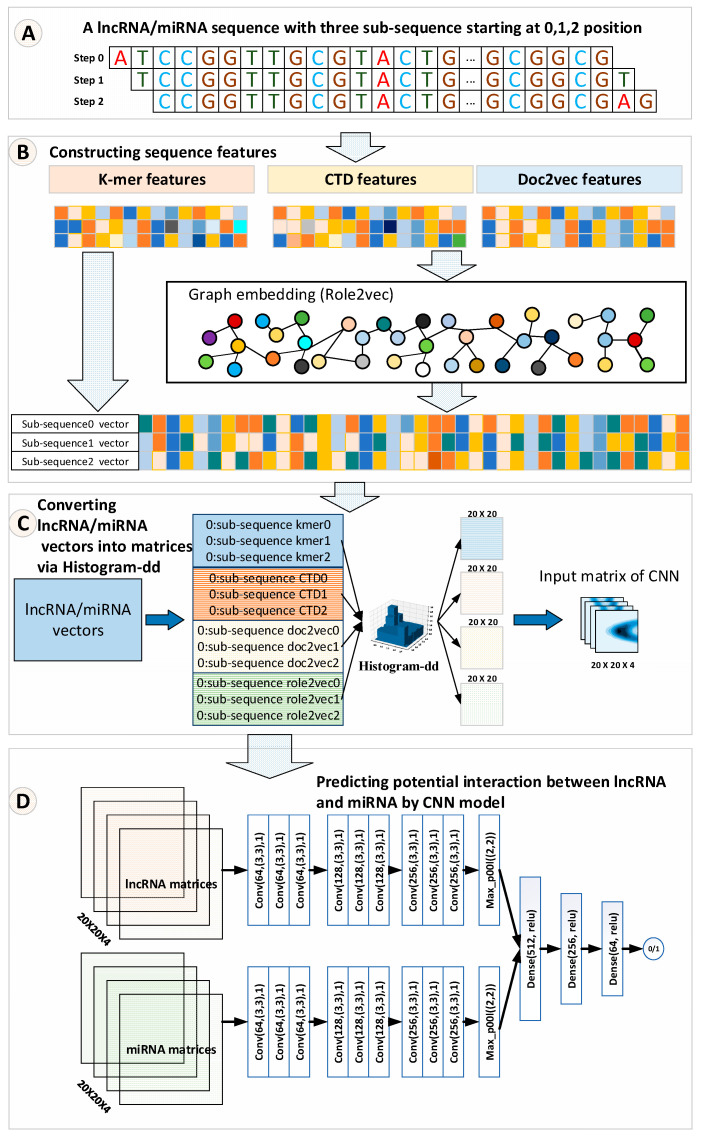
The overall workflow of LncMirNet. (**A**) Sub-sequence of a lncRNA/miRNA starting with position 0, 1, 2 respectively; (**B**) process to construct k-mer, CTD, and doc2vec and graph embedding features; (**C**) process to convert the lncRNA/miRNA vectors into a matrix; (**D**) process to predict potential interaction between lncRNA and miRNA by a CNN model.

**Figure 2 molecules-25-04372-f002:**
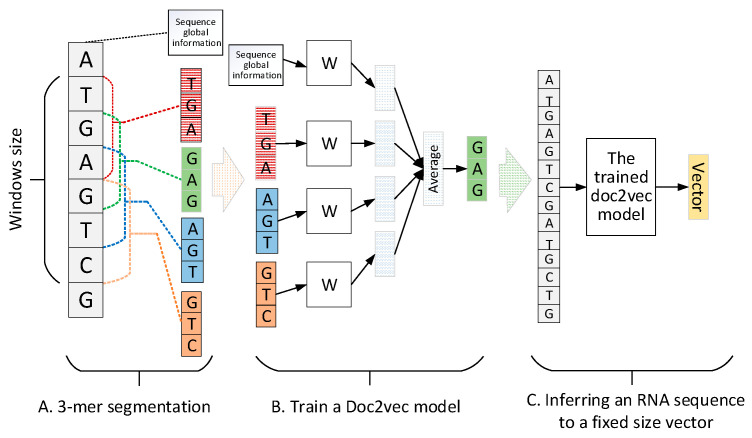
The training and inferring pipeline of doc2vec. (**A**) 3-mer segmentation process; (**B**) training process of a doc2vec model; (**C**) inferring process of doc2vec to encode a RNA sequence to a fixed-size vector.

**Figure 3 molecules-25-04372-f003:**
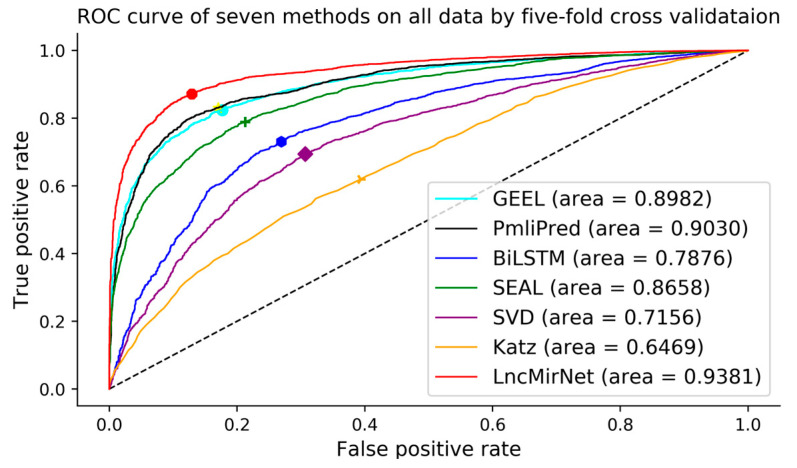
Receiver operating characteristic curves of seven methods on all data by five-fold cross validation.

**Table 1 molecules-25-04372-t001:** Effects of feature information in terms of prediction accuracy.

	k-mer	k-mer, CTD	k-mer, CTD, doc2vec	k-mer, CTD, doc2vec, Graph Embedding
Training	0.8609	0.8802	0.9048	0.9140
Test	0.8004	0.8188	0.8321	0.8534

**Table 2 molecules-25-04372-t002:** The results of the six methods by five-fold cross validation on all data.

	Sensitivity	Specificity	F1-Score	Accuracy	AUC	MCC
GEEL	0.8040	**0.8401**	0.8187	0.8220	0.8982	0.6445
PmliPred	0.8800	0.7118	0.8117	0.7959	0.9030	0.6004
BiLSTM	0.8027	0.6263	0.7239	0.7145	0.7876	0.4359
SEAL	0.7650	0.8097	0.7825	0.7874	0.8658	0.5754
SVD	0.6548	0.6594	0.6595	0.6571	0.7156	0.3142
Katz	0.5969	0.5961	0.5953	0.5964	0.6459	0.1930
LncMirNet	**0.9158**	0.7910	**0.8620**	**0.8534**	**0.9381**	**0.7124**

**Table 3 molecules-25-04372-t003:** Different β in negative sample generation.

β	Number of Positive Samples	Number of Negative Samples	AUC
0.25	15,386	3846	0.8519
0.5	15,386	7693	0.8729
1.0	15,386	15,386	0.9381
2.0	15,386	30,772	0.9067
4.0	15,386	61,544	0.8834

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
