# Peer review of "LncMirNet: Predicting LncRNA–miRNA Interaction Based on Deep Learning of Ribonucleic Acid Sequences"

_molecules, 2020, doi:10.3390/molecules25194372_

Round 1

Reviewer 1 Report

I thought this presentation of LncMirNet was interesting and potentially useful to the community. I felt that the authors made a good attempt at explaining the rationale behind the different methods they used to generate their algorithm.

However, I had a few issues that should be relatively straightforward to address.

  1. While the figures are aesthetically pleasing, I felt that the figure legends didn't describe much of anything. A perfect example of this is 1E - with a bunch of boxes getting smaller and the figure legend saying " is prediction process by CNN". This isn't useful and the figure cannot stand on its own without a legend. I would recommend going through each of the legends and adding sufficient text so that a reader can understand what is going on (maybe someone who isn't intimately aware of lncmirnet?). Similarly, in Figure 2 it isn't clear to me what the box labeled "seq info" is referring to - what kind of information?
  2.  In the text on lines 143-149, the authors describe how CTD is used in examining amino acid sequence and structure, but they do not explain how they translate this to ribonucleotides. This should be explained in an RNA-centric manner.
  3. On line 229 - what is global topic information of an RNA sequence? 
  4. Table 3 should be widened so that the AUC/MCC numbers are not running over one another (and so that the table headers are not overlapping).
  5. The ROC curve in figure 3 is particularly weird. Was only one data point utilized for the LncMirNet? The red lines are two straight lines whereas the other colors are what you typically see for an ROC plot (wavy lines). This doesn't make sense to me and calls into question the accuracy of their model.
  6. Finally, the documentation on their github repository is woefully incomplete. Their model cannot be used by anyone if they do not provide the documentation to support the code. The readme is one line: "LncMirNet is a robust and high confidence computational tool to predict potential interactions between lncRNAs and miRNAs" which does not say anything about installing or running the program. It is standard practice when publishing an algorithm (particularly one that is to be used by the public) to have sufficient documentation that others can test it on their own data.
  7. Which leads me to my last point - without adequate ability to test the algorithm the scientific community (and the reviewers here) have no way of actually ensuring the algorithm works as stated. This should be remedied before publication.

Author Response

1. While the figures are aesthetically pleasing, I felt that the figure legends didn't describe much of anything. A perfect example of this is 1E - with a bunch of boxes getting smaller and the figure legend saying " is prediction process by CNN". This isn't useful and the figure cannot stand on its own without a legend. I would recommend going through each of the legends and adding sufficient text so that a reader can understand what is going on (maybe someone who isn't intimately aware of lncmirnet?). Similarly, in Figure 2 it isn't clear to me what the box labeled "seq info" is referring to - what kind of information?

Thanks for your nice suggestion

Figure 1 and Figure2 is modified by the above suggestion. Figure 1 adds some legend and details CNN model more clear. Moreover, we add a Supplementary Figure 1 to show the detailed structure of the CNN model.

“seq info” is the sequence global information. Doc2vec uses local context information and global information to encode an RNA sequence to a fixed size vector. And in Figure 2, the “seq info” is replaced by “sequence global information”.

====

2. In the text on lines 143-149, the authors describe how CTD is used in examining amino acid sequence and structure, but they do not explain how they translate this to ribonucleotides. This should be explained in an RNA-centric manner.

Thanks for your nice suggestion

We add an example of how to compute CTD features at line 154 and mark it into red.

For example, we use a toy RNA sequence ATACGTACTGCT GACGTAGC which contains five adenines (As), five thymines (Ts), five guanines (Gs), and five cytidines (Cs) to show how to calculate the CTD features. The composition includes four features which are frequency of adenines, thymines, guanines, and cytidines respectively. The toy RNA sequence contains 5 A, 5 T, 5 G, and 5 C, so composition is equal to 5/20 = 0.25, 5/20 = 0.25, 5/20 = 0.25, and 5/20 = 0.25. Transition includes AT, AC, AG, TG, TC, and GC six features which describe the percent frequency with the conversion of four nucleotides between adjacent positions. AT represents the percent frequency of A adjoining T or T adjoining A. AC, AG, TG, TC, and GC are the same formulation of AT. Therefore, the transition for the toy RNA sequence is equal to 2/19 = 0.105, 3/19 = 0.158, 2/19 = 0.105, 4/19 = 0.211, 2/19 = 0.105, 4/19 = 0.211. Distribution is five relative positions along the transcript sequence of each nucleotide, with 0 (first node), 25, 50, 75, 100% (last node), to measure the nucleotide distribution. For As, the 0% is location at first position in toy RNA sequence, 25, 50, 75, and 100% at 3rd, 7th, 14th, 18th position respectively. So, As are 1/20 = 0.05, 3/20 = 0.15, 7/20 = 0.35, 14/20 = 0.7, and 18/20 = 0.9. Likewise, Ts, Gs, and Cs are 0.1, 0.3, 0.45, 0.6, 0.85, 0.25, 0.5, 0.65, 0.8, 0.95, 0.2, 0.4, 0.55, 0.75, 1. We use A0, A1, A2, A3, A4, T0, T1, T2, T3, T4, G0, G1, G2, G3, G4, C0, C1, C2, C3 and C4 to represent the 20 features.

====

3. On line 229 - what is global topic information of an RNA sequence? 

Thanks for your nice suggestion

Global topic information is widely used in NLP (Natural Language Processing). Here, the global topic information represents sequence global information corresponded to local context information. To directly understand, global topic information is replaced by global information.  And it is marked as red at line 252.

====

4. Table 3 should be widened so that the AUC/MCC numbers are not running over one another (and so that the table headers are not overlapping).

Thanks for your nice suggestion

The overlap of AUC/MCC is modified.

====

5. The ROC curve in figure 3 is particularly weird. Was only one data point utilized for the LncMirNet? The red lines are two straight lines whereas the other colors are what you typically see for an ROC plot (wavy lines). This doesn't make sense to me and calls into question the accuracy of their model.

Thanks for your nice suggestion.

We redraw ROC plot by the following steps:

1) we further trim the true dataset. lncRNASNP2 database records true positive lncRNA-miRNA pair. we select positive lncRNA-miRNA pairs when a record in lncRNASNP2 exists ‘hsa-miR’ and ‘ENST’ simultaneously. And, we obtain 258 miRNAs, 1663 lncRNAs and 15,386 validation lncRNA-miRNA paired interactions and their corresponding sequences

2) we used Knuth-Durstenfeld shuffle algorithm to shuffle miRNA set and lncRNA set. And we generate the same amount of negative samples as known positive samples to obtain a balanced dataset.

3) we rerun LncMirNet and six compared method and redraw ROC plot.

The new Figure 3 is distinct.

====

6. Finally, the documentation on their github repository is woefully incomplete. Their model cannot be used by anyone if they do not provide the documentation to support the code. The readme is one line: "LncMirNet is a robust and high confidence computational tool to predict potential interactions between lncRNAs and miRNAs" which does not say anything about installing or running the program. It is standard practice when publishing an algorithm (particularly one that is to be used by the public) to have sufficient documentation that others can test it on their own data.

Thanks for your nice suggestion.

we add a detailed document of how to run LncMirNet. And, add test data.

source code and document at https://github.com/abcair/LncMirNet.

====

7. Which leads me to my last point - without adequate ability to test the algorithm the scientific community (and the reviewers here) have no way of actually ensuring the algorithm works as stated. This should be remedied before publication.

Thanks for your nice suggestion.

we add a detailed document of how to run LncMirNet. And, add test data.

Source code and document at https://github.com/abcair/LncMirNet.

Reviewer 2 Report

In this paper Yang et al present a new tool - LncMirNet - to predict LncRNA-miRNA interactions; the tool is base on Deep Learning of RNA sequences. Overall, the tool is properly described and available via GitHub, and the paper is well written. The subject of the paper occupies the emerging field of  LncRNA-miRNA interactions and therefore deserves attention. I have the following main concerns.

  1. The major contribution of this paper is not sufficiently demonstrated in terms of applicability and novelty. Where do the authors see the application of LncMirNet? In this regard, I would like to see examples of application on new (previously unassessed for LncRNA-miRNA pairs) datasets and exploration of findings.
  2. Related to above, It will be useful to show that findings by LncMirNet have higher enrichment in known interactions from other biological resources, such as GWAS, GTEx, ClinVar, etc.
  3. Instructions on how to use the tool, as well as a toy-set would be very helpful.

Minor:

The presentation needs thorough grammar and typo revision, examples below:

Line 68: “specific” instead of “specificity”?

Line206: “we believe that the candidated lncRNA-miRNA pair exists interaction”, fix to “candidate” and ”interaction exists”.

Line219: Did the authors mean to include sensitivity formula too?

Author Response

1. The major contribution of this paper is not sufficiently demonstrated in terms of applicability and novelty. Where do the authors see the application of LncMirNet? In this regard, I would like to see examples of application on new (previously unassessed for LncRNA-miRNA pairs) datasets and exploration of findings.

Thanks for your nice suggestion.

We are trying to explore the heterogeneous regulatory relationship among mRNA, lncRNA, circRNA, and miRNA in pancreatic cancer via RNA sequence technology. After mapping the RNA sequence to the reference genome, we usually obtain some novel lncRNA and novel miRNA which not annotate in any database. However, we meet a problem to construct heterogeneous regulatory networks. In the heterogeneous regulatory networks, every node represents mRNA/lncRNA/circRNA/miRNA and every edge denotes their interaction. But, we usually do not know the true interaction between novel lncRNA and novel miRNA. Therefore, to help find the potential interaction between these novel lncRNA and novel miRNA, we build the LncMirNet to identify potential interaction between these novel lncRNA and novel miRNA. Moreover, we use the sequence-based feature to avoid the divergence of different quantified methods.

2. Related to above, It will be useful to show that findings by LncMirNet have higher enrichment in known interactions from other biological resources, such as GWAS, GTEx, ClinVar, etc.

Thanks for your nice suggestion.

3. Instructions on how to use the tool, as well as a toy-set would be very helpful.

Thanks for your nice suggestion.

we detail instructions and add a toy-set to guide running the code

https://github.com/abcair/LncMirNet.

The presentation needs thorough grammar and typo revision, examples below:

Line 68: “specific” instead of “specificity”?

Thanks for your nice suggestion.

“specific” is replaced by “specificity” at line 68.

Line206: “we believe that the candidated lncRNA-miRNA pair exists interaction”, fix to “candidate” and ”interaction exists”.

Thanks for your nice suggestion.

"candidated" is replaced by "candidate". "exists interaction" is replaced by "interaction exists" at line 227.

Line219: Did the authors mean to include sensitivity formula too?

Thanks for your nice suggestion.

we add Sensitivity formula at line 238.

Reviewer 3 Report

The manuscript describes a method for prediction of interactions between long-noncoding RNA and microRNA. The positive training set is based on the published lncRNASNP2 database; the negative dataset is constructed from the positive by shuffling of the database sequences; for this step see my question below. According to the described tests, this new method is significantly better, with respect to sensitivity, specificity, accuracy and MCC, than other available methods. The English of the manuscript, however, urgently needs improvement, especially the Abstract.

Question:
line 105: Which shuffling method is used to construct the negative data set? Mono- or dinucleotide shuffling might influence the result especially with respect to their kmer features. For the general idea or problem see the following links:
https://dx.doi.org/10.1093/nar/27.24.4816 ; https://doi.org/10.1093/bioinformatics/16.7.583 ; https://doi.org/10.1093/oxfordjournals.molbev.a040370 ;
https://github.com/wassermanlab/BiasAway/blob/master/altschulEriksonDinuclShuffle.py

Typos, grammar, etc.
=====
line 9: doubled "Correspondence"

line 14: "... resulting in finding the amount of novel lncRNAs and miRNAs, which
contribute to exploring novel regulated patterns." => "... resulting in increasing amounts of novel lncRNAs and miRNAs, which might help to explore novel regulated patterns." Or what is meant here?

line 15: "However, common challenges remain with
lncRNA-miRNA interactions about how to find more valuable information between lncRNAs and miRNAs." I don't understand the meaning of this sentence.

line 15: "However, common challenges remain with lncRNA-miRNA interactions about how to find more valuable information between lncRNAs and miRNAs. Because there are few computational tools available for predicting potential lncRNA-miRNA interaction based on a sequential level." => "However, common challenges remain with lncRNA-miRNA interactions: there are only a few computational tools available for predicting potential lncRNA-miRNA interaction(s?) based on their sequences." Or what is meant here?

line 25: "matrix respectively by histogram2d method friendly for the learning pattern of CNN." => "matrix more friendly for learning patterns by the CNN." Or what is meant here?

line 28: "LncMirNet achieved accuracy and Area Under Curve (AUC) of more than 2%,
respectively, and improved Matthews correlation coefficient (MCC) over 6%." => "LncMirNet increased accuracy and Area Under Curve (AUC) by more than 2%, respectively, over that of the other programs, and improved Matthews correlation coefficient (MCC) by more than 6%."

line 36: "Noncoding" => "noncoding"

line 49: "Zhi An et al." => "Huang et al."

line 52: "Yu An et al. " => "Huang et al."

line 54: "Yu An et al." => "Huang et al."

line 57: "Shuang et al." => "Zhou et al."

line 62: "Qiang et al." => "Kang et al."

line 68: "specificity" => "specific"
"inconsistent in different" => "inconsistent between different"

line 84: "obtain" => "obtained"

line 84: "outperforms" => "outperformed"

line 104: "literatories" => "literature"

line 172: "graph whose construction" => "graph the construction"

line 205: "select sigmoid as an activation function" => "select a sigmoid activation function"

line 214: "16GB RAM" => "16 GB RAM"

Formulas 1--4: Remove the italics (i. e., math mode) from "words" like Specificity, F1-score (this is not F1 minus score), Accuracy, Matthews correlation coefficient.

line 302: "Table 3: the" => "Table 3: The"
Increase column width of this table to give some space around individual headers and numbers in the AUC and MCC columns, respectively.

line 307: "We generate negative samples with same number of know positive samples to obatin balanced dataset." => "We generate the same amount of negative samples as known positive samples to obtain a balanced dataset."

line 308: "positive dataset and negative dataset is" => "positive and negative dataset are"

line 341: "construction of the database" Which database is meant here?

Author Response

Question:
1. line 105: Which shuffling method is used to construct the negative data set? Mono- or dinucleotide shuffling might influence the result especially with respect to their kmer features. For the general idea or problem see the following links:
https://dx.doi.org/10.1093/nar/27.24.4816 ; https://doi.org/10.1093/bioinformatics/16.7.583 ; https://doi.org/10.1093/oxfordjournals.molbev.a040370 ; 
https://github.com/wassermanlab/BiasAway/blob/master/altschulEriksonDinuclShuffle.py

Thanks for your nice suggestion.

we use the Knuth-Durstenfeld shuffle algorithm to shuffle lncRNA set and miRNA set to construct the negative data set.

Typos, grammar, etc.
=====
line 9: doubled "Correspondence"

Thanks for your nice suggestion.

doubled "Correspondence" is modified on line 9.

=====

line 14: "... resulting in finding the amount of novel lncRNAs and miRNAs, which
contribute to exploring novel regulated patterns." => "... resulting in increasing amounts of novel lncRNAs and miRNAs, which might help to explore novel regulated patterns." Or what is meant here?

Thanks for your nice suggestion

The meaning is that the highly increasing development of RNA sequence technology results in numerous novel lncRNAs and miRNAs, which might help to explore novel regulated patterns. And the rewritten sentences are marked in red on line 13.

====

line 15: "However, common challenges remain with
lncRNA-miRNA interactions about how to find more valuable information between lncRNAs and miRNAs." I don't understand the meaning of this sentence.

Thanks for your nice suggestion.

the sentence meaning is that ‘However, the increasing numerous unknown interactions between lncRNAs and miRNAs, which may hinder to find the novel regulated pattern.  The rewritten sentences are marked in red on line 15.

====

line 15: "However, common challenges remain with lncRNA-miRNA interactions about how to find more valuable information between lncRNAs and miRNAs. Because there are few computational tools available for predicting potential lncRNA-miRNA interaction based on a sequential level." => "However, common challenges remain with lncRNA-miRNA interactions: there are only a few computational tools available for predicting potential lncRNA-miRNA interaction(s?) based on their sequences." Or what is meant here?

Thanks for your nice suggestion.

the sentence meaning is that 'However, the increasing numerous unknown interactions between lncRNAs and miRNAs, which may hinder to find the novel regulated pattern. Meanwhile, wet experiment to identify the potential interaction is costly, time-consuming. Furthermore few computational tools are available for predicting potential lncRNA-miRNA interaction based on a sequential level'. The rewritten sentences are marked in red on line 13.

====

line 25: "matrix respectively by histogram2d method friendly for the learning pattern of CNN." => "matrix more friendly for learning patterns by the CNN." Or what is meant here?

Thanks for your nice suggestion.

The sentence meaning is that 'these features are fed into CNN model, which are converted by a histogram-dd method that can fuse multiple types of features into a matrix'. The rewritten sentences are marked in red on line 23.

====

line 28: "LncMirNet achieved accuracy and Area Under Curve (AUC) of more than 2%,
respectively, and improved Matthews correlation coefficient (MCC) over 6%." => "LncMirNet increased accuracy and Area Under Curve (AUC) by more than 2%, respectively, over that of the other programs, and improved Matthews correlation coefficient (MCC) by more than 6%."

Thanks for your nice suggestion.

The sentence'LncMirNet achieved accuracy and Area Under Curve (AUC) of more than 3%, respectively, and improved Matthews correlation coefficient (MCC) over 6%' is rewritten by 'LncMirNet increase accuracy and Area Under Curve (AUC) by more than 3%, respectively, over that of the other programs, and improve Matthews correlation coefficient (MCC) by more than 6%.'. And the rewritten sentences are marked in red on line 27.

====

line 36: "Noncoding" => "noncoding"

Thanks for your nice suggestion.

'Noncoding' is replaced by 'noncoding', and marked as red on line 36 

====

line 49: "Zhi An et al." => "Huang et al."

Thanks for your nice suggestion.

'Zhi An et al.' is replaced by 'Huang et al.' , and marked as red on line 49 

====

line 52: "Yu An et al. " => "Huang et al."

Thanks for your nice suggestion.

'Yu An et al. ' is replaced by 'Huang et al.' , and marked as red on line 52

====

line 54: "Yu An et al." => "Huang et al."

Thanks for your nice suggestion.

'Yu An et al. ' is replaced by 'Huang et al.', and marked as red on line 54.

====

line 57: "Shuang et al." => "Zhou et al." 

Thanks for your nice suggestion.

'Shuang et al.' is replaced by 'Zhou et al.', and marked as red on line 57.

====

line 62: "Qiang et al." => "Kang et al."

Thanks for your nice suggestion.

'Qiang et al.' is replaced by 'Kang et al.', and marked as red on line 64.

====

line 68: "specificity" => "specific"
"inconsistent in different" => "inconsistent between different"

Thanks for your nice suggestion.

"specificity" is replaced by "specific" and 'inconsistent in different' is replaced by 'inconsistent between different' and marked as red on line 68.

====

line 84: "obtain" => "obtained"

Thanks for your nice suggestion.

'obtain' is replaced by 'obtained', and marked as red on line 84.

====

line 84: "outperforms" => "outperformed"

Thanks for your nice suggestion.

'outperforms' is replaced by 'outperformed', and marked as red on line 85.

====

line 104: "literatories" => "literature"

Thanks for your nice suggestion.

'literatories' is replaced by 'literature', and marked as red on line 95.

====

line 172: "graph whose construction" => "graph the construction"

Thanks for your nice suggestion.

'graph whose construction' is replaced by 'graph, the construction', and marked as red on line 193. 

====

line 205: "select sigmoid as an activation function" => "select a sigmoid activation function"

Thanks for your nice suggestion.

'select sigmoid as an activation function' is replaced by 'select a sigmoid activation function', and marked as red on line 226. 

====

line 214: "16GB RAM" => "16 GB RAM"

Thanks for your nice suggestion.

'16GB RAM' is replaced by '16 GB RAM', and marked as red on line 231. 

====

Formulas 1--4: Remove the italics (i. e., math mode) from "words" like Specificity, F1-score (this is not F1 minus score), Accuracy, Matthews correlation coefficient.

Thanks for your nice suggestion.

Formulas 1--4 are removed italics. And 'F1-score' is replaced "F1-socre (F1)", and marked as red on line 235. 

====

line 302: "Table 3: the" => "Table 3: The"
Increase column width of this table to give some space around individual headers and numbers in the AUC and MCC columns, respectively.

Thanks for your nice suggestion.

'Table 3: the' is replaced 'Tabel 2: The'. And the adding space let AUC and MCC columns not overlapped, and marked as red on line 326.

====

line 307: "We generate negative samples with same number of know positive samples to obatin balanced dataset." => "We generate the same amount of negative samples as known positive samples to obtain a balanced dataset."

Thanks for your nice suggestion.

'We generate negative samples with same number of know positive samples to obatin balanced dataset.' is replaced by "We generate the same amount of negative samples as known positive samples to obtain a balanced dataset.", and marked as red on line 331.

====

line 308: "positive dataset and negative dataset is" => "positive and negative dataset are"

Thanks for your nice suggestion.

'positive dataset and negative dataset is' is replaced by "positive and negative dataset are", and marked as red on line 332.

====

line 341: "construction of the database" Which database is meant here?

Thanks for your nice suggestion.

the database is the lncRNA-miRNA interaction database. Owing to the improved accuracy of LncMirNet, it may contribute to constructing the lncRNA-miRNA interaction database, and marked as red on line 365.

Round 2

Reviewer 2 Report

The authors responded to some of my concerns. However, they have not sufficiently addressed major concerns, including demonstration of the usability of LncMirNet (points 1 and 2 of the original review report). In their response they mention applying the tool on newly identified LncRNAs and MiRNAs, but this is not demonstrated in the text of the paper, and not exemplified with new findings that are supported by the current knowledge. Also, I do not see where in the text is point 2 addressed?

Finally (and importantly) the language still needs thorough correction, also for style, repetitions, etc. For example, line 155 and 157 are partially redundant.

Author Response

Thanks for your nice suggestion.

Question 1

The authors responded to some of my concerns. However, they have not sufficiently addressed major concerns, including demonstration of the usability of LncMirNet (points 1 and 2 of the original review report). In their response they mention applying the tool on newly identified LncRNAs and MiRNAs, but this is not demonstrated in the text of the paper, and not exemplified with new findings that are supported by the current knowledge. Also, I do not see where in the text is point 2 addressed?

Answer:

How to use LncMirNet?

There are four steps to use LncMirNet.

  1. prepare lncRNA sequences and miRNA sequences with Fasta format.
  2. using k-mer, CTD, doc2vec, and role2vec to calculate lncRNA sequences features and miRNA sequence features. 
  3. feed these features to the LncMirNet model.
  4. the predicted results are listed in a result file with csv format

On the link https://github.com/abcair/LncMirNet, there is a part to identify interactions between novel lncRNA and novel miRNA. 

Moreover, all parts in the article have been polished carefully including words, sentences, paragraphs and style.
